# Epidural Abscesses as a Complication of Interleukin-6 Inhibitor and Dexamethasone Treatment in a Patient with COVID-19 Pneumonia: A Case Report

**DOI:** 10.3390/medicina59040771

**Published:** 2023-04-16

**Authors:** Valdis Ģībietis

**Affiliations:** 1Pauls Stradiņš Clinical University Hospital, 13 Pilsoņu iela, LV-1002 Riga, Latvia; valdis.gibietis@rsu.lv; 2Department of Internal Diseases, Riga Stradiņš University, 16 Dzirciema iela, LV-1007 Riga, Latvia

**Keywords:** COVID-19, dexamethasone, tocilizumab, epidural abscess, bacteremia

## Abstract

A 66-year-old female patient was hospitalized with severe COVID-19 pneumonia, which led to hypoxia requiring oxygen support with high-flow nasal cannulae. She received anti-inflammatory treatment with a 10-day dexamethasone 6 mg PO course and a single infusion of IL-6 monoclonal antibody tocilizumab 640 mg IV. Treatment led to gradual reduction of oxygen support. However, on Day 10, she was found to have *Staphylococcus aureus* bacteremia with epidural, psoas, and paravertebral abscesses as the source. Targeted history taking revealed a dental procedure for periodontitis 4 weeks prior to hospitalization as the probable source. She received an 11-week antibiotic treatment, which led to resolution of the abscesses. This case report highlights the importance of individual infection risk assessment before the initiation of immunosuppressive treatment for COVID-19 pneumonia.

## 1. Introduction

The interleukin-6 (IL-6) receptor monoclonal antibody tocilizumab is included in the international guidelines for the treatment of COVID-19, as the RECOVERY [1] and REMAP-CAP [2] trials observed that the use of tocilizumab in combination with dexamethasone moderately improves mortality in COVID-19 patients with a severe disease course, worsening condition, increased oxygen demand, and significant inflammatory response [3]. Conflicting data suggest an association of tocilizumab with an increased risk of bacterial infection [4]. This case report discusses a patient with severe COVID-19 pneumonia, who experienced a significant bacterial infection as a complication during treatment.

## 2. Case Report Description

A 66-year-old female patient was acutely hospitalized by the emergency medicine service to a university hospital in Latvia. She complained of persisting fever with body temperature reaching 39 °C for 13 days, fatigue, and progressive dyspnea. Medical history included symptoms of bronchitis with suspicion of asthma during the prior 3 months, grade 2 systemic hypertension, and recently diagnosed type 2 diabetes (glycated hemoglobin 6.8% three months prior). She had been taking perindopril, amlodipine, indapamide, metformin, and inhaled bronchodilators. She reported a history of allergic drug reactions to penicillin group, tetracycline, and gentamycin (angioedema) that had occurred more than a decade earlier. The patient was not vaccinated against COVID-19. The patient was a non-smoker, reported rare alcohol consumption, denied recreational drug use, had no occupational risk factors, and HIV testing was negative.

Objective findings revealed peripheral oxygen saturation (SpO_2_) of 80% in room air, rising to 94% with non-rebreather oxygen mask at flow rate of 12 L per minute. Her respiratory rate was 20 breaths per minute on supplemental oxygen, heart rate 78 beats per minute, and systemic blood pressure 154/78 mmHg. At this point, arterial pO_2_ was 68 mmHg, SaO_2_—96.5%. Nasopharyngeal and oropharyngeal swab was positive for SARS-CoV-2 RNA in PCR testing. Native computed tomography imaging of the thorax revealed signs of advanced bilateral atypical pneumonia in organizing stage consistent with COVID-19 pneumonia (Figure 1). The patient’s body mass index was 33.9 kg/m^2^.

According to the local protocols, based on the United States National Institutes of Health (NIH) guidelines [3], the patient started a 10-day course of dexamethasone 6 mg PO once daily as well as prophylactic dose low-molecular-weight heparin. The patient was hospitalized in respiratory isolation ward in a monitored room. On Day 1, the oxygen support was increased to high-flow nasal cannulae (HFNC), and during Day 2, it was up-titrated to the maximum flow of FiO_2_ 100% at 60 L per minute, which produced SpO_2_ of 86 to 94% depending on the body position. Due to unsatisfactory saturation, additional oxygen support with non-rebreather oxygen mask at 15 L of oxygen per minute was added over the HFNC, sustaining SpO_2_ of 94%. Arterial blood gas analysis showed pO_2_ of 59 mmHg (PaO_2_/FiO_2_ ratio of 59) and SaO_2_ of 92.9%, which fit into the target range of 92–96% for COVID-19 patients [3]; therefore, the oxygen support was not escalated to non-invasive ventilation.

In accordance with local criteria—rapidly worsening COVID-19 pneumonia requiring oxygen support, less than 72 h spent in the hospital, and C-reactive protein (CRP) above 40 mg/L—the patient received a weight-adjusted infusion of 640 mg of IV tocilizumab in a single dose. The criteria were based on RECOVERY [1] and REMAP-CAP [2] trials and NIH guidelines [3]. On Day 3, the patient noted a significant improvement of her subjective state and a decrease in dyspnea. Her body temperature had normalized. She remained on oxygen support through HFNC. The oxygen support was reduced to a simple oxygen mask on Day 10 and gradually decreased until discontinuation on Day 15.

On Day 5, the patient complained of new onset back pain at the level of the 12th thoracic vertebra (Th12) of sharp quality during movements with intensity of 4 to 5 out of 10, which was managed with non-opioid analgesics. On Days 8 and 9, the patient had repeated episodes of febrile body temperature, accompanied by leukocytosis in full blood count on Day 7, although her CRP had dropped from 141.8 to 6.6. mg/L (see Table 1). Due to unexplained febrility and leukocytosis, blood cultures through two peripheral sites were performed on Day 10, with subsequently positive result showing methicillin-sensitive *Staphylococcus aureus* (MSSA) bacteremia (resistant to penicillin G, moderately susceptible to ciprofloxacin, and susceptible to erythromycin, clindamycin, tetracycline, chloramphenicol, linezolid, rifampicin, trimethoprim/sulfamethoxazole, and gentamycin). Because of previously reported allergic reactions to penicillin group antibiotics, the patient started a second-choice treatment with an IV course of vancomycin 1500 mg BID with subsequent serum trough concentration monitoring with a target range of 15 to 20 mcg/mL. Due to back pain associated with unexplained bacteremia, the patient underwent magnetic resonance imaging (MRI) of her thoracic and lumbar spine, which revealed small epidural abscesses within the dorsal epidural space at Th9 to 12 (Figure 2), as well as epidural fat infiltration of 0.4 × 0.8 × 2.2 cm at lumbar vertebra 4–5 (L4–5) and psoas sinister muscle 1.3 × 1.4 × 2.8 cm abscess at L5 level and smaller abscesses in paravertebral muscles at L4–5 level. The patient also had lumbar spondylarthritis, mainly at L4-S1 level, with foraminal stenoses. Testing for alternative causative agents for epidural abscesses, e.g., *Mycobacterium tuberculosis*, was not performed because MSSA was already established as the cause due to positive blood cultures.

Upon a targeted repeated history taking, the patient recalled a dental procedure approximately four weeks prior to hospitalization due to periodontitis. A multidisciplinary decision was made to manage the patient’s spinal infection conservatively without surgical treatment. The patient had no immunologic or vascular phenomena associated with endocarditis. Transesophageal echocardiography did not visualize any bacterial vegetations on heart valves. Due to higher theoretical efficacy of beta-lactam, e.g., oxacillin, nafcillin, rather than vancomycin [5,6], on Day 15, after a directly observed successful slow infusion of oxacillin without any adverse effects, the patient continued treatment with oxacillin 2 g IV every 4 h. The IV antibacterial treatment lasted for a total of 4 weeks, with an additional 7 weeks of outpatient treatment with trimethoprim/sulfamethoxazole 960 mg PO every 8 h. The choice of oral antibiotic was based on the antibiogram described above. At Week 8, a repeated MRI of the spine demonstrated no remaining abscesses and diminished inflammatory changes with remaining edema in paraspinal muscles around the left L4–5 joint. The back pain gradually disappeared. No elevation in body temperature, leukocyte count or CRP was seen. The patient discontinued the antibiotics after 11 weeks of treatment without adverse sequelae.

## 3. Discussion

The presented clinical case occurred during autumn 2021, which, in Latvia, marked the wave of the highest mortality for patients infected with COVID-19 during the pandemic. The mortality peaked at 52 COVID-19-related deaths per 100,000 inhabitants in November 2021, reaching the third highest mortality rate among European Union countries, following Bulgaria and Romania during that month. The higher disease severity of the Delta variant coupled with a low vaccination coverage in the country were important factors to blame for the surge of severely ill hospitalized patients.

This case report demonstrates a patient with severe COVID-19 [3]. RNA sequencing to determine the SARS-CoV-2 variant was not performed in this patient but can be assumed to be Delta because of the 100% Delta (B.1.617.2.) variant prevalence in the Latvian reference laboratory during the period when the patient was diagnosed [7]. The patient had rapid clinical deterioration on Day 13 of illness, which led to hospitalization. The comorbidity of the alleged subacute bronchitis and asthma discouraged her from seeking earlier medical help, as she interpreted the early symptoms as a mere exacerbation, leading to an acute admission at a state of severe hypoxia.

The treatment of hospitalized adults who require supplemental oxygen is based on therapies that directly target SARS-CoV-2 in the early stages and immunosuppressive/anti-inflammatory therapies in later stages to counter the effects of a dysregulated immune/inflammatory response to SARS-CoV-2 leading to tissue damage. In most hospitalized patients who require conventional oxygen, dexamethasone with remdesivir is recommended by the guidelines of NIH [3]. This recommendation is backed by several randomized trials, including ACTT-1 [8] and CATCO [9]. However, in the presented case, the patient was on Day 13 of her illness upon presentation. Remdesivir is not recommended for immunocompetent patients symptomatic for >7 days [10] and was not used. Dexamethasone targets the inflammatory response initiated by SARS-CoV-2 and is recommended in patients with COVID-19 who need supplemental oxygen to meet their prescribed oxygen saturation levels. This recommendation is supported by a meta-analysis of seven randomized trials [11]. The patient received a 10-day course in the standard dose of 6 mg orally.

By the end of Day 1 of hospital stay, the patient was escalated to HFNC oxygen. In such patients, NIH guidelines recommend initiating combined immunomodulator treatment with dexamethasone and oral baricitinib or with dexamethasone and intravenous tocilizumab [3]. Tocilizumab was available at our institution and was administered according to the recommended weight-adjusted dosage. Tocilizumab is a recombinant humanized monoclonal antibody targeted at interleukin-6 (IL-6) receptor. It is approved by the FDA and UK for use in patients with rheumatologic disorders and cytokine release syndrome induced by chimeric antigen receptor T cell therapy [3]. The beneficial effects of this treatment are largely based on the results of the two largest randomized trials—RECOVERY [1] and REMAP-CAP [2]—that demonstrated mortality benefit, including patients who exhibited rapid respiratory decompensation associated with an inflammatory response. The rationale for the use of tocilizumab is its inhibitory effect on interleukin-6, which is one of the key pro-inflammatory cytokines driving the acute inflammatory pneumonic process. The local criteria for administration of tocilizumab in COVID-19 patients also included several contraindications—known hypersensitivity, any other severe infection in addition to COVID-19, hepatic transaminases over 5 times the upper limit of normal, thrombocytopenia < 50 × 10^9^/L, neutropenia < 2 × 10^9^/L, and ongoing active immunosuppressive treatment. At the moment of administration, the patient did not have any data on the existence of these contraindications.

The patient received a combined treatment of two immunomodulatory drugs—dexamethasone and tocilizumab. She demonstrated a gradual clinical improvement and a steady reduction of oxygen support. On Day 7 of the hospital stay, she had an increase in blood leukocyte count with neutrophil predominance, which, at that point, was attributed to corticosteroid treatment; however, on Day 8, she had a repeated episode of febrile body temperature, which was later proven to be due to MSSA bacteremia with epidural and paravertebral abscesses as the source. At presentation, an important anamnestic event was overlooked and not reported by the patient—a dental procedure for periodontitis. Although it had been performed approximately 4 weeks prior to hospitalization, it is a probable source of bacteremia, with spinal infection as a complication exacerbated by the combined immunomodulatory treatment in the context of a severe COVID-19, which may also act as an immunosuppressive factor by itself, reducing the patient’s ability to naturally clear the bacteremia. Diabetes is another well-known risk factor for spinal infections [12].

The duration of antibiotic therapy for epidural abscesses is not well defined. Sources cite 4 to 12 weeks as adequate durations [13]. A relatively long antibiotic duration was chosen in this patient to reduce the risk of recurrence because she was managed conservatively with no drainage of the abscesses [14].

Talamonti et al. [15] reported a case series of six COVID-19 patients with spinal epidural abscesses, indicating an unusually high incidence of this disease in this patient group. Most of the patients did not have typical risk factors for spinal infection, five of them were hypertensive, two were obese, and one had diabetes. The authors hypothesize that the infections may have been caused by the coexistence of an initially asymptomatic bacterial contamination along with COVID-19-induced endotheliitis. Interestingly, three of the patients had received tocilizumab for COVID-19. Sampogna et al. [16] also reported two patients with COVID-19 with respiratory failure requiring mechanical ventilation who acquired spinal epidural abscesses with MSSA and *Enterococcus faecalis* as the causative agents. Both had received tocilizumab, supporting the role of both COVID-19-related and drug-induced immunosuppression along with preexisting clinical factors as the reasons for secondary bacterial infection. Mohamed Ramlee et al. [17] reported three cases of delayed spinal infections following a recent SARS-CoV-2 infection up to three months before the diagnosis. It was not reported whether the patients received any immunosuppressive COVID-19 treatment. Choudhury et al. [18] reported a patient with recurrent and persistent MSSA bacteremia and osteomyelitis, complicated by a spinal epidural abscess, bioprosthetic valve endocarditis, and aortic root abscess despite antibiotic treatment while having tested positive for SARS-CoV-2 infection, implying a COVID-19-induced immunocompromised state with functional exhaustion of CD4 and CD8 T-cells as the potential underlying mechanism for the persistence of such infections. In a report by Chu et al. [19], a 60-year-old patient with diabetes and gingivitis developed *Streptococcus oralis* spinal infection with paraspinal, psoas, and epidural abscesses one week after recovery from a mild COVID-19 infection. The patient had not received any immunosuppressive treatment but did have risk factors for spinal infection—gingivitis and diabetes—similar to the case presented in this report. Our case further strengthens the idea that there is an interplay among initially asymptomatic bacteremia, comorbidities, immune, and endothelial effects of COVID-19 itself along with immunosuppressive treatment, which may explain the development of secondary bacterial infection, particularly spinal epidural abscesses, in such patients.

Similarly, there have been numerous studies and reports on influenza as a predisposing factor for secondary bacterial infection summarized by Radovanovic et al. [20] It is hypothesized that multiple immunological mechanisms may play a role, e.g., damage of the tracheobronchial epithelial layer causing local immunologic response suppression and promotion of bacterial adherence and translocation [21]. As a similar respiratory viral pathogen, SARS-CoV-2 may induce analogous mechanisms for secondary infection.

There is a lack of high-quality trialsthat specifically assess infection risk during treatment with dexamethasone and tocilizumab in COVID-19 patients. In a literature review of 36 studies, Koritala et al. [4] showed mixed results with variable significance for the association of IL-6 inhibitors with risk of infections in patients with COVID-19. Some studies observed an increase in infection risk, mainly bacteremia, while others showed no difference or even a decrease. In a recent retrospective study by Sandhu et al., patients with severe COVID-19 pneumonia treated with tocilizumab experienced high rates of secondary infection—45.5% versus 24.5% in controls [22]. Interestingly, in a study by Kooistra et al. on patients with COVID-19, it was shown that immunomodulatory treatment with dexamethasone and tocilizumab considerably reduced the value of procalcitonin and CRP for detection of secondary infections in COVID-19 patients [23]. This finding may explain why even during and after the onset of back pain and fever, the CRP value remained low while leukocytosis increased in the presented case.

Overall, a careful risk-benefit assessment with thorough history taking is important before initiating a combined immunomodulatory/immunosuppressive treatment in a severe COVID-19 patient to prevent serious adverse effects of the treatment, mainly secondary bacterial infection. The chosen treatment was indicated for the presented patient according to trial data and international guidelines; however, the history of a recent dental procedure due to an oral infection might be a reason to consider alternative approach or a possible dose reduction of the IL-6 inhibitor. Further studies regarding secondary bacterial infection risk in COVID-19 patients treated with immunosuppressive drugs are needed.

## 4. Conclusions

High-flow oxygen therapy and anti-inflammatory therapy with dexamethasone and the IL-6 inhibitor tocilizumab in a patient with severe COVID-19 pneumonia effectively promoted recovery. However, the patient experienced an infectious complication—abscesses in the epidural space and paravertebral muscles. Individual infection risk assessment is required for patients when initiating treatment with immunosuppressive drugs, as both COVID-19 along with immunomodulatory medication may provoke secondary bacterial infection.

## Figures and Tables

**Figure 1 medicina-59-00771-f001:**
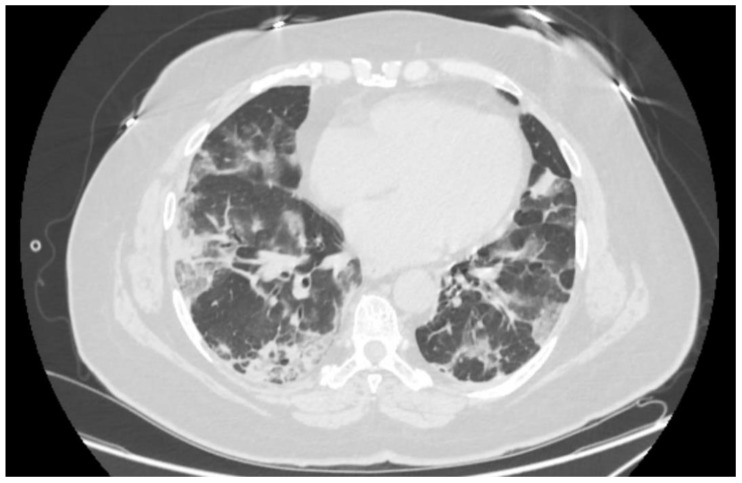
Axial computed tomography image of the lungs on the first day in the hospital.

**Figure 2 medicina-59-00771-f002:**
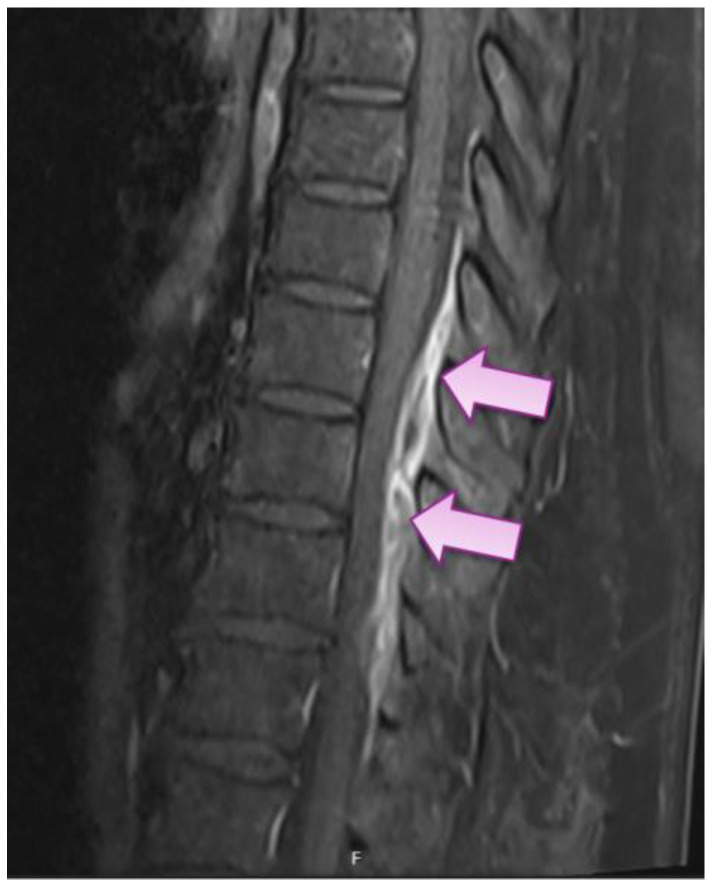
Sagittal magnetic resonance image showing dorsal epidural infiltration with small abscesses from the upper border of Th9 to Th12 (arrows).

**Table 1 medicina-59-00771-t001:** Laboratory findings.

Marker	Day 1	Day 7	Day 15	Unit	Reference Range
Blood leukocytes	6.3	19.6	7.8	×10^9^/L	4–10
Serum C-reactive protein	141.8	6.6	6.8	mg/L	0–5

## Data Availability

Not applicable.

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
