# Peer review of "Epidural Abscesses as a Complication of Interleukin-6 Inhibitor and Dexamethasone Treatment in a Patient with COVID-19 Pneumonia: A Case Report"

_medicina, 2023, doi:10.3390/medicina59040771_

Round 1

Reviewer 1 Report

Interesting case. However, it would be great if you could add a part of literature review on epidural abcesses in Covid-19 and its causes to your discussion. You can use articles like this: 

Spinal epidural abscess in COVID-19 patients by G. Talamonti et al.

Author Response

Thank you for the comment! Discussion has been updated. Lines 175-203.

Reviewer 2 Report

“The higher disease severity of Delta variant coupled with a low vaccination coverage in the country were important factors to blame for the surge of severely ill hospitalized patients”.  Was the sequencing analysis done on samples from these patients to confirm SARS-COV-2 infection y Delta variant.

“In a recent retrospective study by Sandhu et al., patients with severe COVID-19 pneumonia treated with tocilizumab experienced high rates of secondary infection – 45.5% versus 24.5% in controls”

How do doctors and hospitals manage secondary infection in SARS-CoV-2 infected patients, whenever tocilizumab is prescribed/given to patients?

Reviewer 3 Report

This is a well written report of a patient who developed several MSSA abscesses during acute COVID 19 illness. The patient had periodontitis and authors attribute this to be the source of MSSA bacteremia.

The case is well written, case presentation is detailed and well organized. I think that this case will be of value for practicing clinicians, and I would like to see it published. This being said I have some major and minor comments that author should address before publication: 

Major comments: 

1. Line 145-148- mechanism of infection and pathogenesis should be expended here. It is possible that Tocilizumab contributed to development of infection, but I disagree that it was the sole cause. The dental extraction happened 4 weeks before COVID 19 infection and administration of immunosuppressive therapy. Hence, it is possible that patient had low grade protracted bacteremia with seeding to psoas and paravertebral and epidural spaces and following treatment with steroids and tocilizumab infection just got worse. This is one hypothesis. Another possibility is that oral cavity lesion never healed and that during the infection patient had transient bacteremia that would remain transient if she did not receive immunosuppression, in which case, she was unable to clear bacteremia. In any case, I do not think the exact mechanism will be known, but more should be discussed about possibilities regarding pathogenesis.

2. In a recent study it has been described that influenza infection is a risk factor for secondary bacterial infections. I think the same/similar might apply for COVID, so please discuss in more details regarding similarity in influenza and COVID ability to cause secondary bacterial infections: Influenza Myopericarditis and Pericarditis: A Literature Review - PubMed (nih.gov)

3. Conclusion should not be in bullet points but as a paragraph. My take home point for this study is that both COVID 19 infection itself, as well as immunosuppressive therapy can lead to development of secondary bacterial infections and author should re-write this part

Minor points: 

4. Abstract- Staphylococcus aureus should be italicized.

5. The patient was reported to have prediabetes- please report HgbA1c

6. Line 67 and throughout the text pathogen should be written in italics and MSSA abbreviation can be used.

7. Line 69 - please specify dose of Vancomycin?

8. Line 102-105- please avoid repetition, you have demonstrated in case presentation that she had severe disease, so just shorten this to severe disease without further explanation.

9. Line 142- staphylococcal can be changed to MSSA

Round 2

Reviewer 3 Report

I would like to thank the authors on detailed revision of their case report. In the current form the paper is acceptable for publication. 

Author Response

Thank you for your review!

Reviewer 4 Report

I would like to congratulate the authors as the manuscript has improved substantially after the modifications.The authors have sufficiently replied to most of the reviwer's points. However,  I would only like to ask for further clarifications on a few sentences:

4) Why wasn’t non-invasive ventilation used in this case?

SaO2 of 92.9%, which fit into the target range of 92-96%, therefore the oxygen support was not

escalated to non-invasive ventilation. Manuscript updated, lines 59-60.

The authors should further explain what they mean by "target range". Target range for what specifically and according to which authors? Please provide references for this point.

9) Could the authors provide more detailed information about all the physical exam and lab work that was done, including the exclusion of immunologic and vascular phenomena associated with endocarditis, if blood cultures were collected through two peripheral sites, antibiogram results including sensitivity to other drugs, exclusion of HIV and tuberculosis for example.

HIV testing was negative. Blood cultures were collected through two peripheral sites. The patient had no immunologic and vascular phenomena associated with endocarditis. These facts and susceptibility data are now added to the manuscript. TBC was not tested.

Why was the patient not tested for TB? If this test was not performed a proper disclosure should be included as TB could be cause of epidural abscesses.

Author Response

Thank you for your thorough review!

The authors should further explain what they mean by "target range". Target range for what specifically and according to which authors? Please provide references for this point.

This is based on NIH guidelines for COVID-19 patients who receive supplemental oxygen that state: “a target SpO2 of 92% to 96% seems logical, considering that indirect evidence from patients without COVID-19 suggests that an SpO2 of <92% or >96% may be harmful.” Reference added.

Why was the patient not tested for TB? If this test was not performed a proper disclosure should be included as TB could be cause of epidural abscesses.

TB was not tested because the first finding, which prompted MRI of the spine, was the positive blood culture for S. aureus. Since the causative agent was already established, we did not test for other causative pathogens. I have now disclosed this in the manuscript, line 88-90.